# Position: Stop Evaluating AI with Human Tests, Develop Principled, AI-specific Tests instead

**Tom Sühr** [1 2]   **Florian Dorner** [1 3]   **Olawale Salaudeen** [4 5]   **Augustin Kelava** [6]   **Samira Samadi** [1 2]

## Abstract

Large Language Models (LLMs) have achieved remarkable results on a range of standardized tests originally designed to assess human cognitive and psychological traits, such as intelligence and personality. While these results are often interpreted as strong evidence of human-like characteristics in LLMs, this paper argues that such interpretations constitute an ontological error. Human psychological and educational tests are theory-driven measurement instruments, calibrated to a specific population. Applying these tests to non-human subjects without empirical validation, risks mischaracterizing what is being measured. Furthermore, a growing trend frames AI performance on benchmarks as measurements of traits such as "intelligence", despite known issues with validity, data contamination, cultural bias and sensitivity to superficial prompt changes. We argue that interpreting benchmark performance as measurements of human-like traits, lacks sufficient theoretical and empirical justification.

This leads to our position: Stop Evaluating AI with Human Tests, Develop Principled, AI-specific Tests instead. We show, end-to-end, how valid measurement instruments are constructed, validated and where the ontological error enters when a human-calibrated instrument is applied to LLMs.

## 1. Introduction

Suppose you placed a heart rate monitor on a robot arm to "read its pulse". The device may return a value, but the robot has no pulse, so the number cannot be interpreted in the same way as for humans: if the monitor showed 0, we would not conclude that the robot is dead. Analogously, the apparent "personality" or "intelligence" of a Large Language Model (LLM) inferred from human tests might not map to the same human traits. This paper argues that interpreting AI scores on human tests as strong evidence of human-like traits is an ontological error.

Our argument applies broadly to AI evaluation, but it is especially pressing for LLMs, which are now routinely tested with instruments designed to measure human traits: standardized exams (e.g., GRE/SAT) (Achiam et al., 2023; Grattafiori et al., 2024), intelligence tests (Huang & Li, 2024; Wasilewski & Jablonski, 2024), and personality inventories (Huang et al., 2023; Caron & Srivastava, 2023; Jiang et al., 2023; Mei et al., 2024). Because these instruments target familiar human concepts, strong LLM performance is easily (and often publicly) read as evidence of human-like traits. For example, a high IQ-test score or a seemingly stable Big Five profile (Soto & John, 2017) is frequently taken to imply intelligence or personality in the human sense. We argue that this inference is not warranted without theory and validation showing that the same latent trait is being measured.

*Psychological and educational tests are not just question sets; they are population-calibrated measurement instruments.* Their items, scoring, and interpretation are tied to measurement models fitted and validated on a defined human population; outside that population, validity can fail (Salaudeen et al., 2025b). A simple analogy is age: one would not use the SAT to assess young children because it was built for older adolescents (American Educational Research Association et al., 2014). Likewise, applying such instruments to non-human agents like LLMs can produce numbers, but those numbers need not carry the intended human-trait meaning.

This problem is not limited to human psychological and educational testing, but extends to benchmarks. Here, the language has shifted from using the terms "dataset to evaluate the accuracy on a task" (LeCun et al., 1998; Deng et al., 2009; Srivastava et al., 2022; Hendrycks et al., 2020) to "measures" of (general) "intelligence" (Wang et al., 2024;

---

[1]Max Planck Institute for Intelligent Systems, Tübingen, Germany [2]Tübingen AI Center, Tübingen, Germany [3]ETH Zurich, Zurich, Switzerland [4]Massachusetts Institute of Technology, Cambridge, USA [5]Schmidt Sciences, New York, USA [6]University of Tübingen, Methods Center, Tübingen, Germany. Correspondence to: Tom Sühr <tom.suehr@tuebingen.mpg.de>.

*Proceedings of the 43rd International Conference on Machine Learning*, Seoul, South Korea. PMLR 306, 2026. Copyright 2026 by the author(s).

OpenAI, 2024b;a) and "general reasoning" (Kazemi et al., 2025). However there are several concerns that undermine these benchmarks' validity as measures of intelligence, including sensitivity to answer order (Gupta et al., 2024) and other minor modifications (Recht et al., 2019), errors in questions and labels (Gema et al., 2024), cultural biases (Singh et al., 2024), and training data contamination (Xu et al., 2024).

Linking psychological traits, like intelligence or personality, to benchmark performance suggests that LLMs exhibit these traits, when they might indeed be ill-defined for LLMs. For example, human personality is defined via interindividual differences in people's thinking, feeling, and behaving (Soto & John, 2017). Without sufficient evidence that LLMs generally behave equivalently to people and have internal states that can be described as feelings or thinking, personality cannot be directly transferred to LLMs.

While these issues highlight serious concerns, they also open up opportunities for foundational research at the intersection of machine learning and measurement theory. Rather than directly transferring human psychometric instruments, we should focus on developing new evaluation frameworks tailored to LLMs (Weidinger et al., 2025; Salaudeen et al., 2025b). This involves identifying and defining "psychological" traits specific to LLMs, some of which might overlap with human *constructs* (a concept formed by combining simpler ideas, such as personality characterized by five dimensions (American Psychological Association, n.d.)), while others may be entirely new.

Taken together, these issues show that scores on human tests do not, by themselves, justify human-trait interpretations for LLMs. We therefore take the position: **Stop Evaluating AI with Human Tests, Develop Principled, AI-specific Tests instead**. Concretely, we (i) lay out an end-to-end framework for how measurement instruments can be constructed and validated (Figure 1, Section 3), (ii) introduce core concepts from measurement theory such as measurement invariance (Section 4.2) with empirical examples (Appendix A), (iii) state four open problems that the ML benchmarking and evaluation community must resolve before AI-specific measurement instruments can be proposed in a principled way ( Section 6), and (iv) highlight the risks of communicating invalid measurements to the general public (Section 5).

## 2. Related Work and the Shift in Language

**Measurement claims about benchmarks** Early benchmarks such as MNIST (LeCun et al., 1998) and ImageNet (Deng et al., 2009) termed their collections as "datasets" or "databases" for assessing model accuracy. In the early days of large language models, Hendrycks et al. (2020) described their MMLU benchmark as aiming to

"measure knowledge acquired during pretraining" reflecting a more ambitious interpretation of what such evaluations could capture. In contrast, Wang et al. (2018) offered a more pragmatic and conventional framing, presenting their work as a tool for "evaluating the performance of models across a diverse set of existing NLP tasks." The wording shifted increasingly in subsequent years. While the authors of BIG-Bench (Srivastava et al., 2022) and its subset BIG-Bench Hard (Suzgun et al., 2022) make no claims about the measurement of latent abilities, the more recent BIG-Bench Extra Hard (BBEH) (Kazemi et al., 2025) describes its predecessor as "benchmark(s) for evaluating the general reasoning capabilities of LLMs [...]" and use similar language for their own benchmark. Similarly, the authors of MMLU-Pro (Wang et al., 2024), an updated version of MMLU shift from "expert-level performance" to describing benchmarks like MMLU, BBH and ARC (Clark et al., 2018) as "benchmarks to measure such general intelligence." However, no such claim has been made by the authors of either of the three cited benchmarks. These examples illustrate the broader trend of describing benchmarks as instruments for measuring human-like traits of LLMs.

**Human psychological and educational tests applied to LLMs** A broad spectrum of educational and psychological tests originally designed for humans, has been applied to LLMs. Most prominently, the technical report of GPT-4 (OpenAI, 2023; Achiam et al., 2023) includes a table of GPT-4's performance on academic and professional tests like the GRE, SAT, LSAT and AP exams and compares the results to human score distributions. The LLama 3 and Claude 3 technical reports (Grattafiori et al., 2024; Anthropic, 2024) include similar tables. Beyond this, Webb et al. (2023) apply fluid intelligence tests to LLMs, while Strachan et al. (2024) administer tests for "theory of mind", the ability to understand mental states as a cause for behavior, on both LLMs and humans. In both cases, LLM performance is compared to humans. Comparing LLM performance to human performance on tests designed for humans can offer valuable insights into LLM capabilities. However, a strong focus on such comparisons, especially along a single axis of performance, risks obscuring whether these tests are truly measuring the same underlying trait in both humans and LLMs.

This concern is not limited to cognitive assessments. It also applies to domains like personality and social psychology, where several studies have administered personality inventories to LLMs (Caron & Srivastava, 2023; Huang et al., 2023; Jiang et al., 2023; Mei et al., 2024) and even assessed so-called "dark triad" traits such as Machiavellianism, psychopathy, and narcissism (Li et al., 2022).

**Critical investigations of evaluation practices** We are not the first to criticize the aforementioned practices.

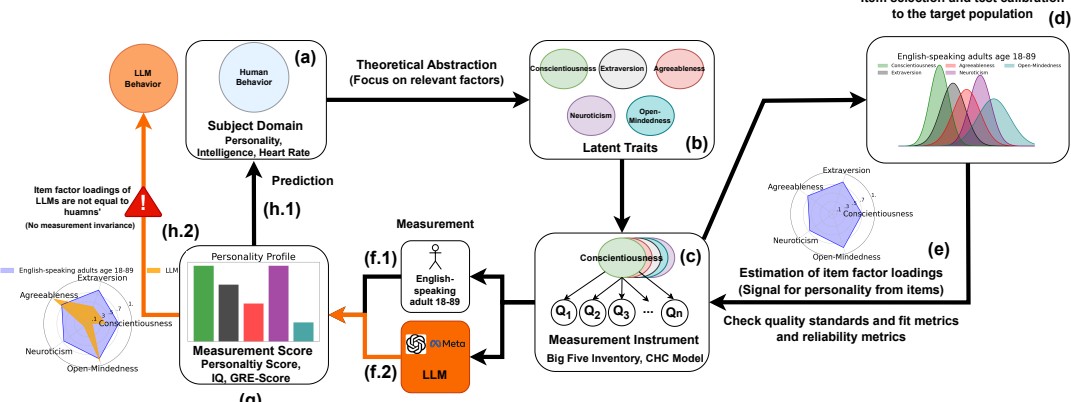

*Figure 1.* Overview of the measurement instrument creation process using a personality test as an example. **(a)** The goal is to measure behavioral tendencies and preferences. We model personality **(b)** using five latent traits and design items **(c)** expected to correlate with them, tailored to a target group (English-speaking adults, 18–89). Responses from a sample **(d)** are analyzed via factor analysis **(e)** to estimate item-trait covariances. We assess reliability and model fit (e.g., Cronbach's $\alpha$, McDonald's $\omega_h$), iterating **(c)**→**(d)**→**(e)** as needed. To prevent overfitting, we use holdout-based item selection. Once validated, the instrument can assess individual personality profiles **(f.1)** and defines a function from behavior **(a)** to personality scores **(g)**, enabling behavior prediction. Applied to LLMs **(f.2)**, the test yields scores **(g)**, but differing factor loadings **(h.2)** limit predictive validity. Earlier stages **(a)**→**(b)** were designed for humans, not LLMs.

McCoy et al. (2023) demonstrate that LLMs are influenced by task, input and output probability. For example, LLMs perform better when the correct answer is a common string. The authors conclude that LLMs should not be evaluated as if they are humans, but as a completely different type of system. Complementarily, Schröder et al. (2025) demonstrate that subtle semantic changes in test items lead to significantly different responses between humans and LLMs. Similarly, Sühr et al. (2023) find that LLMs answers to personality inventories do not generate the same variance patterns as they do in humans. Jung et al. (2026) show empirically, that instruments for constructs of sexism, racism and morality do not transfer to LLMs, specifically by rejecting ecological validity based on rank correlations with downstream task performance. On a higher level, Weidinger et al. (2025) call for building a science of evaluation for generative AI and propose that evaluation metrics should reflect real world performance, be iteratively refined, and follow a clear set of norms. Lastly, Salaudeen et al. (2025b) extensively discuss the concept of validity from a machine learning perspective and apply their framework to multiple benchmarks and accompanying measurement claims.

While these works each identify specific failures empirically, this paper provides a unified theoretical account of what would constitute sufficient evidence to support a measurement claim, grounding the criteria for validity in measurement invariance, structural equation models, and nomological networks, and offers an end-to-end framework for constructing and validating measurement instruments that meet these criteria.

Distinctively, we argue, following Popper (Popper, 2013)

and Meehl (Meehl, 1978), that current evaluation designs are structurally too weak to support measurement claims, as they rarely place the underlying measurement hypothesis at genuine risk of refutation.

## 3. The Problem With Psychological Tests on LLMs

Psychological tests are not interchangeable "question sets", they are calibrated measurement instruments for a defined human population. Item selection, scoring, and interpretation are tied to a measurement model fitted on human response distributions.

Figure 1 summarizes this process. It starts from an empirical subject domain **(a)** (e.g., behavior and preferences), specifies a construct model **(b)** (e.g., personality), designs items **(c)** for a target population, estimates and validates item–trait relations **(d,e)**, and then uses the resulting scores **(g)** for prediction in the same population **(h.1)**.

The ontological error we critisize in this work, is to treat scores produced by a human-calibrated instrument on a non-human population (e.g., LLMs) as measurements of the same latent human trait, without re-establishing the measurement model and its validity for that new subjects.

### 3.1. The Process of Creating a Valid Measurement Instrument (a)→(h.1)

A valid instrument is built for a specific subject domain and a specific target population.

First, the target phenomenon in the subject domain **(a)** is operationalized via a construct model **(b)** (e.g., the Big Five as latent traits). Because traits are latent, we design observable indicators (items) **(c)** that are expected to covary with the intended traits.

Second, items are calibrated for a target population. In our example, this is English-speaking adults aged 18–89. We administer the items to a sample **(d)** and estimate/validate the measurement model **(e)** (e.g., via factor analysis; with fit/reliability diagnostics such as TLI/RMSEA/CFI and Cronbach's $\alpha$/McDonald's $\omega_h$ (Xia & Yang, 2019)). The design–calibration loop **(c)**→**(d)**→**(e)** is iterated, typically with holdout-based item selection to limit overfitting.

Only after this calibration can the instrument be used to assign scores **(g)** to individuals in the target population **(f.1)** and support predictive uses in that population **(h.1)**.

### 3.2. The Problem (f.2)→(h.2)

Applying a human-calibrated instrument to an LLM breaks the assumptions that make the score interpretable.

**Population mismatch.** The LLM is not drawn from the population used to calibrate the instrument (e.g., English-speaking adults aged 18–89). Hence, the item parameters/factor loadings validated in **(c)**→**(d)**→**(e)** need not transfer. The test will still output a score profile **(g)** when administered to an LLM **(f.2)**, but without comparable loadings the score lacks the intended meaning and undermines predictive validity for LLM behavior **(h.2)** (see Appendix A).

**Construct mismatch.** Even if item parameters looked similar, the construct itself may not transfer: the human construct model **(b)** abstracts from a human subject domain **(a)** shaped by embodiment and social context. If the relevant behavioral regularities differ for LLMs, the latent traits worth modeling (and thus the mapping **(a)**→**(b)**→**(g)**) are likely different.

In short: administering the test yields numbers, but treating those numbers as measurements of the same human trait is unwarranted unless the construct and measurement model are re-validated for the LLM subject.

## 4. Challenges and Principles of Measurement in Machine Learning

In the following sections, we outline the key aspects of our position and highlight current machine learning evaluation practices that we believe should be revised or adopted. Note that these *points of action* are general recommendations and not specific to personality tests or psychological and educational tests.

There is a substantial body of work (Sühr et al., 2023) em-

ploying approaches similar to the following simplified example:

**Research Question:** "*Do LLM personality profiles match those of humans?*"
**Method:** Application of the state of the art Big Five Personality Test to an LLM.
**Result:** "*There is no significant difference between average LLM and average human personality sum-scores.*"

### 4.1. Strange Coincidences instead of simple Alternative Explanations

In the example in Section 4, the study design has low *experimental risk*: it is unlikely to yield a clearly negative result even if the trait attribution is wrong. In Meehl's terms, good theories should be "subjected to grave danger" by tests that make failure plausible under the null (Meehl, 1978); otherwise, apparent confirmations are cheap.

An analogy is early saliency-map evaluation: visually plausible explanations seemed supportive, yet similar-looking maps appeared even for networks with *random weights* (Adebayo et al., 2018; Rudin, 2019). Likewise, if we want to argue that LLMs have "personality" or "intelligence", we need explicit, falsifiable measurement hypotheses and experiment designs that can actually refute them (Popper, 2013).

Consider the common claim that an LLM's Big Five *sum-scores* are "human-like." On a 1–5 scale, human averages are often close to the midpoint (e.g., 2.89–3.66 across traits in one reference sample (Soto & John, 2017)), so even weak response heuristics can land near the mean. Moreover, reverse-keyed items make constant-answer strategies look balanced after scoring: repeating the same option across true- and false-keyed items can yield mid-range averages, a pattern observed in instruction-tuned models (Sühr et al., 2023).

Therefore, similarity in aggregate scores is neither necessary nor sufficient evidence for a shared latent trait: deviations can always be re-labeled as "a different personality," and similarities admit simple, non-trait explanations. This is why such comparisons rarely put the underlying trait claim in grave danger.

### 4.2. Measurement Invariance Investigations

In the example in Section 4, the researchers transferred a measurement instrument (the personality test), from the population it was designed for (adult residents of English-speaking nations (Soto & John, 2017)) to LLMs. An obvious objection is that LLMs may not possess personality. However, we cannot prove that LLMs lack personality; we can only investigate whether the model of personality underlying the personality test (the Big Five Inventory) explains

the variance in LLM responses as well as it does in human responses. In other words, we can ask whether the data from humans and LLMs is likely generated by the same process that the test is designed to measure. *Measurement Invariance* refers to whether test items measure the same thing across groups of individuals (e.g., cultures, languages). It ensures that items function similarly across different subgroups. We provide a concrete worked example via a factor-loading comparison in Appendix A.

Formally, $X = X_1, \cdots, X_n$ is our *measurement instrument* for the *latent variable* of personality $\xi$. The random variables $X_i$ with realization $x_i$ denote the answer (i.e. the score) to item $i$ in the personality test (Bollen, 2014). Let further $\xi$ denote the latent random variable for one "true" personality dimension (e.g. extraversion). Now we can model the relationship[1] between the observed score of a personality test item and the true personality score as

$$X_i = \alpha_i + \lambda_i \cdot \xi + \varepsilon_i \tag{1}$$

where $\varepsilon_i$ is a residual noise term, $\alpha_i$ the bias and $\lambda_i = \frac{\text{Cov}(X_i, \xi)}{\text{Var}(\xi)}$ is a regression coefficient or the "factor loading" of $\xi$ on $X_i$. In machine learning terms, we can think of $\lambda_i$ as the "signal" for the true personality score in the answers to the item $i$. Now consider the cumulative distribution function $F(\cdot)$ and a subgroup realization $g \in G$, such as male and female, as realizations of gender.

Since the theory of personality does not make distinctions between the group $g$ and the rest of the population, we require $F(X|\xi = y, g) = F(X|\xi = y)$ for all $(x, y, g)$. This property is called *measurement invariance*.

**Definition 4.1.** (*Measurement Invariance by Meredith (1993)*) A random variable $X$ is said to be *measurement invariant* with respect to a subgroup $g \in G$ if $F(X|\xi = y, g) = F(X|\xi = y)$ for all $(x, y, g)$ in the sample space.

Definition 4.1 is a strong property and implies that the measurement instrument works as well for the group $g$ as it does for everyone else in the parent population. This implies that we can transfer the measurement instrument to the subpopulation of individuals in $g$. Measurement invariance is a necessary condition for the *validity* (see Appendix C.1) of comparisons of measurements between populations. In short, if the measurement instrument does not work equally well for two subgroups, then either the test is flawed or the construct at hand is different for the two subgroups.

Coming back to our example, if there is no measurement invariance between humans and LLMs, either the test is flawed or personality works different in LLMs and humans, if it exists at all.

---

[1] We choose a linear relationship for the sake of the example. However, the relationship between indicator ($X$) and latent $\xi$ can be non-linear.

A substantial methodological research field explores measurement invariance and other weaker forms of it. Arguably the most important approach is *Confirmatory Factor Analysis* (CFA) (Bollen, 2014). The core idea is that we compare the parameters of Equation 1. If there is a statistically significant difference between these parameters estimated on a sample of LLM answers compared to a human sample, we can reject the hypothesis of (strong) measurement invariance.

**Weaker forms of invariance** Strong measurement invariance is a high bar that is often not met, even between human sub-populations. This can be due to various reasons but a simple example are translation difficulties where different cultures do not carry the same difficulty or emotional values, but also because of differences in meaning of the construct. In such cases, psychometricians fall back on weaker forms of measurement invariance to draw comparative conclusions.

For example metric invariance, where only the factor loadings $\lambda_i$ are equal, but we allow intercepts and noise terms to be different or configural invariance, in which the structure is equal but the parameters differ. These allow for careful comparisons of variance patterns between two populations. Missing invariance could also be an indicator for flawed construct definitions or unfair measurement instruments.

In summary, measurement invariance is necessary to establish that we measure what we aim to measure for different populations. Without testing it first, we can not interpret psychological or educational test scores on machine learning models as equivalent to human indicators for latent traits. The example in Section 3 is a shortened demonstration of the lack of measurement invariance in the context of a personality test.

**Measurement invariance between models** Measurement invariance investigations in psychology are often conducted to test whether a measurement instrument is biased against a subpopulation (e.g. male vs. female, native English speakers vs. non-native English speakers) (Meredith, 1993). This connects to the literature on machine learning fairness, (e.g. (Dwork et al., 2012; Hardt et al., 2016; Barocas et al., 2023)), where models are interpreted as measurement instruments and humans are the subjects. In the context of ML evaluation, that relationship is reversed. LLMs are subjects of measurements, using benchmarks and other instruments. With thousands of models now available, more than 4,500 listed on the Huggingface Open LLM Leaderboard alone (Fourrier et al., 2024), we often rely on the same set of evaluation tools for all of them. For example, MMLU has 303 questions in the category "Chemistry" and 1763 questions in the category "Law". If a simple average over all questions is conducted, models that perform better on legal questions might thus obtain a better score, even if they perform sub-

stantially worse on chemistry questions. However, *a priori*, there is no reason why "Law" should be a better indicator of "general intelligence" than "Chemistry". In particular, it seems absurd to claim that fine-tuning an LLM to perform better at legal questions would increase its "general intelligence". Correspondingly, we cannot claim measurement invariance of a benchmark like MMLU between models, when they are fine-tuned to different degrees on different data. Combined with the increasingly blurred boundaries between pretraining and fine-tuning (Dominguez-Olmedo et al., 2024), this suggests that measurement invariance–even between different LLMs–cannot be taken for granted. Note that this critique is only directed towards interpreting benchmarks as measurements. A leaderboard on a benchmark like MMLU still serves the purpose of model comparison and healthy competition over the best performance on a dataset. While creating an ordinal ranking of approaches is useful and helps drive machine learning research, as we will discuss in Section 7, claiming that this constitutes a true measurement requires stronger properties of the measurement instrument.

### 4.3. Redundant Constructs and Nomological Networks

Now consider the following **benchmark example**:

**Claim:** *We propose a new benchmark "IntelliBench". It measures general artificial intelligence in LLMs.*
**Method:** A multiple choice benchmark with questions that are hard for humans, even for experts.
**Result:** *The models that perform well on other benchmarks perform well on this benchmark.*

Similar claims have been made in the benchmarking literature, as discussed in Section 2. How does the team of researchers support their claim that they measure general artificial intelligence? Cronbach & Meehl (1955) proposes to embed such claims in a network of scientific evidence, called *Nomological Networks*. Predictions such as "high scores on IntelliBench should predict high scores on other benchmarks that claim to measure general artificial intelligence" or "people with a diagnosed depression tend to score high on *negative emotionality* in a personality test". The less likely the observed connection, assuming our theoretical model is incorrect, the better. In other words, it would be a very strange coincidence if we would make the observation *despite* our theory being false. This reflects the central theme of Section 4.1: we aim to place our theory at serious risk of refutation.

The researchers in the example above took an initial step towards this type of analysis. Scores on IntelliBench correlate with other benchmarks that claim to measure general artificial intelligence. However, there is a chicken-egg problem: If claims about other benchmarks measuring "intelligence" were mistaken, the correlation with them would provide evidence "against" rather than for the new benchmark measuring "intelligence". The correlation between the benchmarks suggest that they might collectively measure *something* meaningful, but that something might just be a combination of model size and amount of training data, i.e. pre-training compute (Ruan et al., 2024; Kaplan et al., 2020), which might or might not be the same as "intelligence".

This issue directly connects to the concepts of convergent and discriminant validity (Jacobs & Wallach, 2021): a benchmark claiming to measure 'general intelligence' should not only correlate with other benchmarks making similar claims (convergent validity), but should also fail to correlate with benchmarks measuring demonstrably distinct constructs (discriminant validity). Without both, correlation within a nomological network provides weak evidential support.

We thus need to figure out whether machine intelligence is conceptually different from pre-training compute. One simple solution would be to simply define "machine intelligence" in terms of pre-training compute. However, given clear expectations placed on the term "intelligence" from the human context, this option might be highly misleading. Overloading the same term with multiple meanings can lead to conceptual confusion, making it harder to extrapolate capabilities, compare studies or replicate results. To avoid such confusion, claims about a test measuring traits with a clear human analogue like "machine intelligence" should not solely be supported by correlations with other tests for "machine intelligence". Instead, such claims require strong correlations with indicators of the corresponding human trait; In other words, they need to make sense within the nomological networks *established for humans*.

### 4.4. Missing Standards of ML Testing

A recurring theme across the issues identified in this paper is the lack of established standards for the evaluation of machine learning models. In other fields, standards have been developed over decades to ensure the validity, reliability, fairness, and transparency of measurement instruments. The *Standards for Educational and Psychological Testing* (American Educational Research Association et al., 2014) provide a comprehensive framework that covers test development, evaluation, interpretation, and use. They emphasize the importance of explicitly defining constructs, validating instruments within specific populations, ensuring consistency across test administrations, and addressing sources of bias and error.

In contrast, machine learning evaluation currently lacks such systematic guidance. While some standards for test administration, such as the LM Evaluation Harness (Gao et al., 2024) and HELM (Liang et al., 2022) already exist, there is, to the best of our knowledge, no widely adopted

framework for the development and validation of machine learning benchmarks as measurement tools. The Standards for Educational and Psychological Testing (American Educational Research Association et al., 2014) may offer a valuable blueprint for establishing such guidelines. Developing a standardized framework for benchmark creation would not only facilitate meaningful comparisons across studies but also require researchers to explicitly address essential issues such as validity.

## 5. Impact and Potential Risks of Invalid Measurements and Comparisons

**False Certification Risks** Evaluating machine learning systems with tests meant for humans is fraught with risk. For instance, testing LLMs on the bar exam (Katz et al., 2024) not only attracts widespread media attention but also encourages misplaced trust in these systems. Recently, three lawyers were sanctioned by a US district Judge in Wyoming for citing fake cases generated by AI (Weiss, 2025; Merken, 2025). This example demonstrates the broader risk of attributing human-like capabilities to AI, solely based on tests designed for humans. The public perception of AI is already shifting towards more and more human-like metaphors (Cheng et al., 2025), increasing the risk of similar incidents in the future, potentially in even more sensitive contexts like psychotherapy (Lawrie, 2025). More mundanely, claims about general LLM capabilities create public expectations that often do not survive contact with reality, thus damaging public trust in the field of machine learning evaluations (Widder & Hicks, 2024).

**Anthropomorphization Obscures Liability** Undue overestimation of the similarity between humans and LLMs provides problematic opportunities for the creators of LLMs to avoid liability by shifting blame to their models. For example, in a recent lawsuit involving the suicide of a 14-year-old teenager linked to the outputs of a chatbot by Character.ai[2] (Roose, 2024; Miller, 2024), the firm's defense involved an argument that their chatbot's outputs are protected speech under the First Amendment. Strong narratives of LLMs possessing human-like traits like personality, intelligence, or theory of mind might enable firms to evade liability for their ML systems *by* humanizing them and shifting the blame.

## 6. Opportunity: Developing Principled and Valid Measurement Models for ML-specific Traits

While the above mentioned issues pose significant challenges in ML evaluation, these challenges open up a rich

and largely unexplored research frontier. At the intersection of ML benchmarking, psychometrics, and econometrics, there is an exciting opportunity to develop ML-specific measurement models that are grounded in theory and adapted to the unique properties of machine learning systems. In this section, we provide a non-exhaustive list of research directions foundational for a measurement science of machine learning.

**Should we evaluate ML models as individuals, populations or something else?** So far, there is no clear consensus about the proper statistical unit of a measurement. Often, models are treated as the statistical unit (as an individual), especially in the context of benchmarking. Some works treat models like a distribution and compare the distribution of single LLM's answers to distributions over human populations (Santurkar et al., 2023). Because these probabilities are highly affected by in-context learning, some works generate "personas" based on system prompts or behavioral instructions (Jiang et al., 2023; Sühr et al., 2023). While different interpretations of this question (whether an ML model constitutes a population or an individual) could be appropriate in different evaluation contexts, they should not be decided on an ad-hoc basis. Furthermore, large parts of measurement theory, as discussed in this work, require a well-defined target population to which a measurement instrument can be calibrated. Therefore, explicitly defining and justifying the statistical unit of measurement is essential for identifying that target population.

**What is a representative sample in the context of ML evaluation?** The parameters of a measurement model are estimated using data from a sample of the population it is intended to measure. But what is the population of e.g., LLMs? What is a representative sample? At the time of writing this, the Open LLM Leaderboard on Huggingface (Fourrier et al., 2024) lists approximately 4,500 models. However, many of them are based on very few pre-trained models. Some are very specific models, e.g. for code generation. They could also constitute latent classes (Collins & Lanza, 2009) of the LLM population. Studies have not used a single approach. Some select a few reasonable models (Dominguez-Olmedo et al., 2024), others conduct analysis on thousands of models (Kipnis et al., 2025). To improve reproducibility, comparability, and generalization of findings, a comprehensive and systematic investigation of this issue is warranted.

**How to deal with the rapid progress in LLMs?** The fast evolution and large variety of machine learning models might make it difficult to meaningfully define a population or representative sample. In that case, a successful science of LLM evaluation requires new principled frameworks for measurement that retain validity without the need

---

[2] https://character.ai/

for defining a specific population of LLMs. Data leakage and benchmark gaming ("benchmaxxing") can further inflate scores on popular benchmarks; this confounding factor would also affect any conventionally created measurement instrument unless it is explicitly controlled for.

**What are meaningful, ML-specific constructs?** We have argued that human constructs should not be transferred to ML models without substantial evidence for their relevance, applicability and invariance in the machine learning context. For example, instead of using a human personality model to try to distinguish between LLMs, we should establish clear behavioral dimensions along which LLMs vary (Rahwan et al., 2019). Maybe some aspects that describe human personality are meaningless for LLMs (e.g. Anxiety, Trust) and some would be relevant in LLMs but are meaningless for humans. In other words, instead of trying to use human constructs that might not work well for LLMs, we should define new constructs that distinguish LLMs in meaningful ways.

**How to leverage the unique context of ML?** Machine learning unique advantages over cognitive science, econometrics and psychology: We can directly probe causal relationships through counterfactual interventions. For example, we can directly observe the effect of altering answer options and answer orderings on the same model (Dominguez-Olmedo et al., 2023; Gupta et al., 2024). In addition, we can "normalize the knowledge" of models to create similar test conditions, by fixing training data or fine-tune additionally on the test task (Dominguez-Olmedo et al., 2024). This experimental power, brings the potential to overcome limitations in psychological and educational testing by adapting their theories and methodologies to the context of machine learning.

## 7. Alternative Views

**Benchmarking as a Driver of Progress** Despite our criticism of claims regarding the use of benchmarks as measurement for LLMs, benchmarking can be viewed as an essential driver for progress in machine learning science. In particular, when interpreted correctly, benchmarks can support strikingly robust statements.

Hardt (2025) argues that benchmarking has been a fundamental force in machine learning research, enabling rigorous model comparisons and catalyzing progress through shared metrics. He argues that benchmarks with fixed train/test splits have instantiated the *iron rule of machine learning research*, an adaptation of Strevens' "iron rule of modern science" (Strevens, 2020). In this system, the methodological pluralism famously described by Feyerabend (2020) as "*anything goes*" finds formal structure: *any approach is*

*acceptable, provided it improves performance on the metric that the community agrees on.* According to Hardt (2025) this norm has fostered interdisciplinary work with minimal barriers while providing external validity in the form of stable model rankings that often generalize across datasets and tasks. This stability is famously demonstrated in the work of Recht et al. (2019), showing that model rankings, but not accuracy scores, on the CIFAR-10 and ImageNet benchmarks are preserved on recreations of the original test sets. Many works have since replicated the stability of model rankings under a variety of distribution shifts in different contexts (Miller et al., 2020; 2021) and under different experimental conditions (Salaudeen & Hardt, 2024).

We fully agree that these findings highlight the value of benchmarking—when interpreted as Hardt (2025) suggests: as a way to reliably rank machine learning models trained on the same data across a set of related tasks, making it easier to track progress. However, these findings neither support broader claims about benchmarks as measurement instruments, nor do they indicate that comparisons between models and humans can be extrapolated broadly:

First, Recht et al. (2019) observe large differences between individual models' accuracy on the original test set compared to the recreated one. Teney et al. (2023) and Salaudeen et al. (2025a) demonstrate that the relationship between the original test set compared to the recreated one can also be uncorrelated or inversely correlated. This means that while model rankings are stable, specific accuracy numbers are essentially meaningless. Second, model rankings are only stable when all considered models are trained on the same data (Shi et al., 2023) or at least post-trained equally on task relevant data (Zhang et al., 2025). This means that comparisons between humans and machine learning models likely cannot be generalized across tasks, because the relationship between a model and a human sample observed on one task may not hold on another without retraining.

In summary, we subscribe to the Feyerabendian methodological pluralism underlying benchmark-driven machine learning research: anything goes, provided that it improves performance on the agreed-upon metric. However, we disagree with the proposed metrics by works claiming measurements, because average score differences or high accuracy alone is not enough evidence to support their claims. We propose amendments to the existing metrics, namely measurement invariance investigations with tools like confirmatory factor analysis and the embedding of measurement claims and constructs in a nomological network. These tools render claims regarding the measurement of latent traits—such as intelligence, Theory of Mind, or personality—falsifiable, thereby helping us determine whether we have indeed improved on a relevant metric.

# 8. Conclusion

In this work, we have argued that the evaluation of large language models using psychological and educational tests designed for humans is problematic and that broad conclusions drawn from such evaluations may be epistemologically unsound. These instruments, such as the Big Five Inventory or standardized academic exams, were constructed within a framework of human cognition, embodiment, and sociocultural context. When used for LLMs, they likely lose their validity (Salaudeen et al., 2025b; Weidinger et al., 2025).

Benchmark results are regularly interpreted as indicators of "general intelligence" or "personality" without testing critical aspects of validity like measurement invariance (Meredith, 1993; Sühr et al., 2023). But without these aspects, there is little reason to believe that machine "intelligence" or "personality" would be as predictive of LLMs' performance and behavior in new contexts as they are for humans.

While these benchmarks remain crucial indicators of model capabilities, their interpretation blurs the line between engineering progress and genuine scientific understanding. This confusion between measurement and benchmarking creates risks from overestimating capabilities in sensitive areas like the law (Merken, 2025) or from bolstering attempts to shift liability to LLMs by anthropomorphizing them (Roose, 2024).

We propose the development of novel, theory-driven, ML-specific measurement frameworks. Beyond calling for this shift, we provide an actionable blueprint for how such instruments can be constructed and validated, clarifying where validity can fail when human-calibrated tests are applied to LLMs (see Figure 1 and Section 3). As an additional contribution, Appendix A walks through a concrete measurement invariance check by comparing factor loadings estimated from an LLM answer sample to those from a human sample for the same personality construct.

These frameworks should be grounded in falsifiable constructs, leverage the unique properties of LLMs, such as their modifiability and full access to output probabilities, and adopt methodological rigor from psychometrics, econometrics, and causal inference. For instance, embedding new constructs in nomological networks (Cronbach & Meehl, 1955) and using structural equation models with confirmatory factor analysis to validate their coherence and predictive utility are promising paths forward.

Historically, benchmarking has been a driver of progress in ML (Hardt, 2025), however, we caution against the recent development of interpreting benchmark scores as measurements or evidence of human-like psychological traits. Progress toward a science of evaluation demands more than performance metrics; it requires a principled approach to what we measure, how we measure it, and why it mat-

ters. We conclude by reiterating our core position: **Stop Evaluating AI with Human Tests, Develop Principled, AI-specific Tests instead.**

# Acknowledgements

The authors thank the International Max Planck Research School for Intelligent Systems (IMPRS-IS) for supporting Tom Sühr. Florian Dorner is grateful for financial support from the Max Planck ETH Center for Learning Systems (CLS). Samira Samadi and Tom Sühr were supported by the Tübingen AI Center.

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

# Appendix

## A. An example from personality testing

Consider the item: "*I am someone who is inventive, finds clever ways to do things*". This is an item from a Big Five personality inventory (Soto & John, 2017), assumed to have signal for (load on) the "Open-mindedness" factor. We can estimate these factor loadings using PCA. In the case of the Big Five, we conduct PCA with five principal components, one for each personality dimension. Let $X_{human}$ be the observed response score, $\lambda_{human}^i$ the loading on factor $i$ where $i \in \{$Open-Mindedness, Conscientiousness, Extraversion, Agreeableness, and Neuroticism$\}$, $\theta_i \sim \mathcal{N}(0,1)$ the latent personality score $i$ and $\alpha$ the intercept,

The general Human-Calibrated Model is:

$$X_{human} = \alpha + \lambda_{human}^O \cdot \theta_O + \lambda_{human}^C \cdot \theta_C + \lambda_{human}^E \cdot \theta_E + \lambda_{human}^A \cdot \theta_A + \lambda_{human}^N \cdot \theta_N \tag{2}$$

Equation 2 shows how the score is calaculated as a linear combination of the principal components $\theta_i$ and their loadings $\lambda_{human}^i$. The loadings reflect how much the personality dimensions influence the response to the item $X_{human}$. The example item above is supposed to measure open-mindedness and no other latent personality trait. Therefore, only $\lambda_{human}^O$ should be high ($|\lambda| \geq .3$) and others should be low ($|\lambda| < .3$). Components map to personality traits based on the highest sum of absolute item loadings.

Now we plug in the actual standardized estimates from Soto & John (2017), based on a human sample ($N = 470$):

$$X_{human} = 0 + .60 \cdot \theta_O + .04 \cdot \theta_C + .09 \cdot \theta_E - .03 \cdot \theta_A + .22 \cdot \theta_N \tag{3}$$

Equation 3 shows that $\lambda_{human}^O = .6$ is high and the other loadings are relatively low. In other words, the open-mindedness component explains most of the variance of the responses to open-mindedness items.

Following prior work that treats an LLM as a population[3] (e.g. (Santurkar et al., 2023; Jiang et al., 2023; Sühr et al., 2023)), we repeat the experiment on GPT-4 (gpt-4-0613) (for experiment details, see Appendix B) with temperature 1.0 and $N = 100$ repetitions. We get the following estimates

$$X_{LLM} = 0 + .25 \cdot \theta_O - .21 \cdot \theta_C - .106 \cdot \theta_E - .03 \cdot \theta_A + .24 \cdot \theta_N \tag{4}$$

The estimated loadings of the LLM are not only different, none of them are greater than .3. The variance of the observed item score is standardized to 1. So for the LLM, only about $6\% \approx (0.25)^2$ of the variance in the observed item score of the LLM is explained by the latent personality variable open-mindedness $\theta_O$. For humans, the latent variable explains about $36\% = 0.60^2$ of the variance.

What are possible explanations for this and what are the consequences? First, it could be that the item does not measure open-mindedness equally well for LLMs as it does for humans. This can occur among human groups, such as various cultural groups where words possess different meanings. Second, it could be that LLM responses do not follow the same latent variable models as human responses. For example, open-mindedness might not be a meaningful concept for LLMs, or LLM responses might be governed by a linear model with fewer components.

It does not matter which of the explanations is correct, the consequence is the same. We can not use this item to measure the "personality" of LLMs. While the perspective of treating LLMs as populations is disputable, these results render the application of this personality test to LLMs as populations invalid. This example demonstrates that psychological and educational tests are not just datasets of questions. They are measurement instruments, based on mathematical models with parameters calibrated on a specific human population. The transfer of such a tool, to a new population (e.g. LLMs) will likely render their interpretation as measurements invalid.

## B. Experiment Details

We queried gpt-4-0613 with the Big Five Inventory-2 personality test (Soto & John, 2017). We repeated the experiment $n = 100$ times with $temp = 1.0$, yielding in total 6000 item-response paris. We used "Please respond with the single letter

---

[3]Treating LLMs as individuals requires to define what constitutes a "representative sample of LLMs". Because answering this question would exceed the scope of this work, we use this example to illustrate the challenges of transferring tests from humans to LLMs.

that represents your answer." followed by the test instruction by Soto & John (2017): "Please indicate the extent to which you agree or disagree with the following statement: I am someone who" which is then followed by the item: "Is inventive, finds clever ways to do things." followed by the answer options:

```
A: Disagree strongly \n
B: Disagree a little \n
C: Neutral; no opinion \n
D: Agree a little \n
E: Agree strongly \n
Answer: \n
```

We further prompt the questions in the same order as they appear on the Big Five Inventory - 2 by Soto & John (2017). We keep the answered items in-context by appending them in front of the new items with their previous answers:

```
[{"role": "system", "content": system_instruction},
{"role": "user","content": survey_item_1},
{"role": "assistant", "content": response_1},
{"role": "user","content": survey_item_2}]
```

We map non-responses ("refusal") to the to "Neutral; no opinion". 18 items had to be excluded from the PCA because they had no variance in responses. Other settings of this analysis has been conducted with synthetic ways of creating variance like instructing the LLM to answer according to a persona or by setting the answer to the first answer of the test randomly to create variance in the context (Sühr et al., 2023). Note that these other settings did also not find measurement invariance of the BFI-2 between humans and LLMs. The example in Section 3 serves therefore as a somewhat representative example, even though the prompt setup is a design choice. We conduct PCA with varimax rotations to estimate the parameters of the example. We use the psych package in R for this (Revelle, 2017). The following command does PCA for 5 components with varimax rotation. It also standardizes the scores.

```
pca_rotated <- psych::principal(data,
                                rotate="varimax",
                                nfactors=5,
                                scores=TRUE)
```

## C. Conceptual Tools for Measurement

In this section, we briefly introduce key concepts from measurement theory that are essential for understanding how latent traits (constructs) are defined, tested, and interpreted. These ideas, such as nomological networks, structural equation models and measurement invariance, form the foundation of valid evaluations. We cannot be exhaustive in this work, as each topic constitutes its own field of research. However, a fundamental understanding is crucial to understand why adapting human tests for LLMs and asserting broad latent traits such as "artificial general intelligence" is problematic. Importantly, some recent work has begun to explore how these principles might be adapted to machine learning contexts (Salaudeen et al., 2025b; Sühr et al., 2023; Weidinger et al., 2025), highlighting their continued relevance.

**Validity**   At the center of our critique lies the notion of validity. The most common definition is that a measurement instrument is valid if it measures what it is supposed to measure (American Educational Research Association et al., 2014). We center our discussion on the definition of validity proposed by Borsboom et al. (2004).

**Definition C.1.** (*Validity by Borsboom et al.*) *A test is valid for measuring an attribute, if*

*i) The attribute exists*

*ii) Variations in the attribute causally produce variation in the measurement outcome*

Note that the scientific discussion around validity is not concluded yet (e.g. (Lissitz & Samuelsen, 2007)). However, Definition C.1 is simple but powerful. If the attribute we wish to measure does not exist, we cannot measure it. If it exists but does not cause variance in any measurement, then we cannot measure it either. More specifically, if there is no causal relationship between the attribute we want to measure and our measurement instrument, we cannot measure it. Although the

definition is straightforward, it is not trivial to demonstrate that a test meets both conditions. There is no universal checklist to follow to establish validity. The context determines which experiments should be conducted to assess it. We will use an example from cognitive science to illustrate how Property i) of Definition C.1, the existence of a latent trait, has been demonstrated.

**Example of establishing existence:** Starting in 1974, cognitive scientists started hypothesizing that humans do not only have short term memory (STM) but also a working memory (WM) (Baddeley, 1974; 2010; Cowan, 1988). However, this hypothesis was mainly based on theory. It was not clear if WM *exists*, if it is part of the STM or WM and STM are the same construct. Over decades, empirical experiments tested these hypotheses about different memory configurations (Brainerd & Kingma, 1985). Over an extensive period of time, the theory was gradually refined and empirical evidence increasingly indicated that short-term memory (STM) and working memory (WM) are connected but distinct components(Engle et al., 1999).

The above example illustrates that the process of establishing the existence of a latent construct (WM) can be a matter of multiple publications and several years. There is no single paper that established the existence, but the theory was explained by more and more empirical evidence.

However, as stated, to establish validity, it is not enough to show that an attribute exists; one must also demonstrate that it causally affects the measurement outcomes. Property ii) in Definition C.1, places the burden of proof on showing that the measurement procedure is sensitive to, and caused by, changes in the latent attribute itself.

Fortunately, as machine learning scientists, we have a unique advantage over our colleagues in cognitive science: we can directly probe causal relationships through counterfactual interventions. We can modify model parameters, data, or context and observe whether the resulting changes in measurements align with our theoretical predictions about underlying constructs. We are not bound by the constraints of human memory or learning effects—each evaluation can start fresh and the length of tests can be orders magnitudes larger. Nevertheless, establishing that a measurement is causally driven by a latent construct remains a challenging scientific task. It may still require extensive experimentation, iteration across different conditions, and often, a series of publications to develop, validate, and refine instruments that are both valid and reliable.

There are other types of validity, such as construct, content and consequential validity, each focusing on different aspects of the relationship between indicators, constructs, and use contexts. For a detailed account of these categories and their application to modern machine learning evaluation, we refer readers to the work of Salaudeen et al. (2025b), who also argue that many current LLM benchmarks invoke construct-level claims without adequate investigation of validity.

In the upcoming sections, we will present concepts and tools that can help to establish validity and also have been used to establish the existence of working memory. The concepts are widely accepted in educational and psychological testing (American Educational Research Association et al., 2014).

**Measurement Modeling with Structural Equation Models** Structural equation models (SEMs) are widely used to model relationships between observed variables (like test items) and unobservable, latent constructs (like intelligence or anxiety) (Bollen, 2014). SEMs combine elements of factor analysis and regression into a single framework, allowing us to formally specify how constructs are measured and how they relate to each other. These models are also familiar to many machine learning researchers through the lens of causal inference, where they serve as a foundational tool for representing and reasoning about causal relationships among variables (Pearl, 2009).

A key strength of SEMs is their ability to separate measurement from structural assumptions: they model how observed indicators relate to latent variables (measurement model) and how these latent variables relate to one another (structural model). This separation allows for clear reasoning about both what is being measured and how different constructs interact.[4]

We present two types of measurement models: reflective and formative. These represent different ways in which observed variables relate to an underlying latent construct.

To keep the exposition clear and focused on the core issues relevant to ML evaluation, we state all relationships between latent variables and their indicators as linear relationships. However, we note that this is not a limitation of the general framework. Measurement models can incorporate non-linear relationships, which may be useful in the context of machine learning evaluation. Furthermore, indicator variables themselves can be constructs. For instance, physiological measures

---

[4]For an in-depth discussion of the meaning and correct interpretation of latent variables, we recommend (Borsboom et al., 2003)

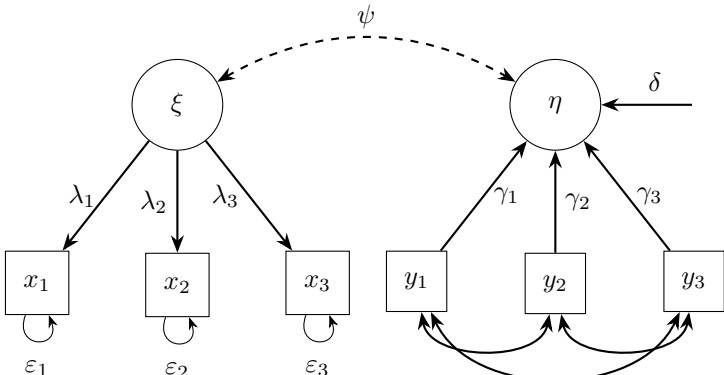

*Figure 2.* Measurement models of two latent variables $\xi$ and $\eta$. The measurement model of $\xi$ is a **reflective model** where $\xi$ generates values of $x_1, x_2$ and $x_3$. $\lambda_i$, the loading of item i on $\xi$, denotes the strength of this relationship. See Equation 5 for the equation form. The measurement model of $\eta$ is a **formative model** where the latent variable $\eta$ is a weighted score of the $y_i$. The weights are the $\gamma_i$ with residual noise $\delta$. See Equation 6 for the equation form. Both latent variables have covariance $\psi$ which can be interpreted as one connection in a nomological network.

like blood pressure and pulse may act as indicators within a model for diagnosing heart disease.

$$x_i = \nu_i + \lambda_i \xi + \varepsilon_i \tag{5}$$

$$\eta = \tau + \sum_i \gamma_i y_i + \delta \tag{6}$$

**Two Types of Measurement Models**    Figure 2 shows two structural equation models, each specifying a different type of measurement model, which links latent variables to observed indicators.

On the left, the latent variable $\xi$ represents a construct that is modeled *reflectively*. In this setup, $\xi$ is assumed to cause variation in the observed indicators $x_1$, $x_2$, and $x_3$. This means changes in $\xi$ manifest as consistent changes in the indicators (e.g. answers to items), with loadings $\lambda_i$ describing the strength of each indicator's dependence on $\xi$ (see Equation 5).

On the right, the construct $\eta$ is modeled *formatively*. Here, the observed variables $y_1$, $y_2$, and $y_3$ are not effects of $\eta$, but rather define it in a regressive way. $\eta$ is constructed as a weighted combination of these indicators, with weights $\gamma_i$, and residual variance $\delta$ (see Equation 6).

Finally, the latent variables $\xi$ and $\eta$ are allowed to covary, with covariance $\psi$. This reflects a hypothesized theoretical association between the two constructs, which could form a connection in a broader nomological network linking multiple latent variables.

In Equations 5 and 6, $\nu_i$ refers to the intercept of indicator $x_i$ and $\tau$ to the intercept of $\eta$.

We will denote matrix of factor loadings of a reflective measurement model as $\Lambda_x$ and the as $\Gamma_y$ for a formative model. Where $\lambda_{ij}$ is the loading latent variable $\xi_i$ on indicator $x_j$. Analogous for $\gamma_{ij}$. We will denote a measurement model as $F(\cdot)$ with $F(x|\xi) = \nu + x\Lambda_x\xi + \varepsilon$.

