# OpenReview forum: "Position: Stop Evaluating AI with Human Tests, Develop Principled, AI-specific Tests instead"
_ICML.cc/2026/Position_Paper_Track — ICML 2026 Position Paper Track regular_

### Official Review · Reviewer_YeVz · 2026-03-10

**Significance:** 3
**Argument Clarity:** 3
**Rating:** 4
**Confidence:** 3

**Questions:**

1. Appendix A demonstrates that the BFI-2 lacks measurement invariance when applied to GPT-4. Did the authors conduct similar analyses on other tests, such as MMLU and IQ tests? Is the absence of measurement invariance a widespread phenomenon, or is it specific to certain tests?

2. Section 6 suggests that ML-specific constructs should be defined. Does the author have any preliminary candidate constructs to propose? For example, could concepts like “consistency,” “calibration,” and “distribution sensitivity” of models serve as substitutes for human “personality” constructs?

3. Before a new framework is developed, how should the community properly utilize existing benchmarks? Does the author recommend completely discontinuing the reporting of human test scores, or merely ceasing to interpret them as measures of human traits?

**Alternative Views Section:**

Yes

**Compliance With Llm Reviewing Policy A Conservative:**

Affirmed.

**Discussion Potential:**

3

**Final Justification:**

The paper presents a clear and timely perspective on an important issue in current LLM evaluation, and introduces relevant psychometric concepts that are often overlooked in the ML community. Its main contribution lies in highlighting the gap between benchmarking and valid measurement, and in providing a principled framework for thinking about this problem.

However, the empirical evidence remains limited, and the proposed future directions are still relatively abstract without concrete operationalization. As a result, while the paper is conceptually valuable and has strong discussion potential, it is not yet fully convincing in its current form. I therefore maintain my borderline accept rating.

**Paper Summary:**

This paper argues that using human-designed psychological and educational tests, such as IQ tests, personality scales, and standardized exams, to evaluate LLMs poses fundamental problems. The authors point out that this constitutes an "ontological error": human tests are measurement tools calibrated for specific populations, and their item selection, scoring, and interpretation are all tied to measurement models that have been fitted to human populations. Directly applying these tools to non-human entities (LLMs) without empirical validation leads to misrepresentations of the constructs being measured.

The main contributions of this paper include:
(1) Proposing an end-to-end framework for the construction and validation of measurement tools;
(2) Introducing core concepts from psychometric theory, such as measurement invariance, and providing empirical examples;
(3) Identifying four open questions that the ML benchmarking community needs to address before developing AI-specific measurement tools;
(4) Highlighting the risks of communicating invalid measurement results to the public.

The authors call on the machine learning community to stop interpreting benchmark scores as measures of human traits and instead develop theoretical frameworks tailored specifically to the unique properties of ML models.

**Position:**

Yes

**Position In Title:**

Yes

**Related Work:**

3

**Strengths And Weaknesses:**

Strengths:

1. The paper presents a well-defined stance on a hot topic in current LLM evaluation. As models such as GPT-4 and Claude 3 frequently demonstrate human-like performance in technical reports, this discussion is highly timely and relevant to the ICML community.

2. The paper introduces core psychometric concepts (measurement invariance, structural equation modeling, nomological networks) and explains why these concepts have been overlooked in ML evaluation. Appendix A provides specific examples of factor-loading comparisons, illustrating the absence of measurement invariance between humans and LLMs.

3. Section 5 discusses in detail the potential risks of invalid measurement, including the risk of false certification (e.g., lawyers sanctioned for AI-generated fake cases) and the risk of liability avoidance (e.g., companies evading responsibility by anthropomorphizing AI). These real-world examples significantly bolster the persuasiveness of the argument.

Weaknesses:

1. Appendix A only presents factor-loading analyses for individual items using a small sample size (N=100 repetitions). While sufficient as an illustrative example, a more comprehensive analysis across multiple models and tests would be more convincing.

2. Although the paper calls for developing AI-specific tests, it offers few details on what specific constructs should be included in the new framework or how these constructs should be operationalized. The research directions proposed in Section 6 are promising but lack initial solutions.

3. Section 7 acknowledges the value of benchmarks as ranking tools, yet the boundary between "benchmarking" and "measurement" sometimes remains unclear. More explicit criteria could be established to determine when a benchmark can be considered a measurement and when it cannot.

**Support:**

3

---

> ### Author Rebuttal · Authors · 2026-03-30
>
> We thank Reviewer YeVz for their careful review and constructive questions.
>
> **W1-Scope of Appendix A:** We agree that Appendix A is only illustrative. Importantly, the paper's central argument can not be that we comprehensively refuted all human-test applications empirically. We point out that works making measurement claims have not provided the evidence required to support the validation of them. The burden of proof lies with those asserting that a human-calibrated instrument measures the same construct in LLMs. Our work provides the framework (Figure 1) and the theoretical tools (measurement invariance and nomological networks) to evaluate this. As we discuss in Section 2, several works we cite provide additional empirical evidence: Sühr et al. (2023) find no measurement invariance across multiple personality tests and models; Schröder et al. (2025) show subtle semantic changes produce systematically different responses in LLMs vs. humans; McCoy et al. (2023) demonstrate LLM responses are driven by output probability rather than the intended construct; and Jung et al. (2026) reject ecological validity for instruments measuring sexism, racism, and morality. We will clarify in the revision that Appendix A sits within this broader picture.
>
> **W2-AI-specific constructs:** Many AI-specific constructs or instruments may ultimately be useful, but only if their measurement properties, external correlates, and validation procedures are examined rigorously for which we provide the necessary tools. We do not offer initial solutions in the form of underspecified alternative constructs as the history of intelligence and personality research suggests that construct development is iterative and often unfolds over many years and multiple publications.
>
> **W3-Boundary between benchmarking and measurement:** We agree this could be stated more clearly. As discussed in our response to Reviewer SCr2 and bYyX, we will add a dedicated sentence to the introduction to clarify the benchmarking/measurement distinction.
>
> **Q1-Measurement invariance beyond BFI-2:** We have not conducted the same analysis for MMLU or IQ tests. For MMLU this is partly because it is ``just'' a dataset: unlike the BFI-2, it does not come with an explicit structural measurement model. There are works that did exploratory factor analysis or PCA on MMLU like Burnell et al. (2023) [1] and their results are not compatible with current human theories of intelligence. IQ tests would indeed be an interesting future direction. More broadly, our point is that claims to measure llm-specific or human-like traits require the validation steps in Figure 1, including measurement checks for cross-population comparisons. Otherwise, benchmark scores are task-performance evidence, not measurements of a shared latent trait.
>
> Measurement invariance should not be assumed before it is established quantitatively. Measurement invariance is a strong requirement but necessary for valid score comparisons. Weaker forms of invariance exist and still support limited comparisons. We will be happy to include weaker forms of invariance in the revision.
>
> **Q2-Preliminary ML-specific constructs:** As stated above, we want to be cautious about proposing new constructs without empirical grounding. In this area, a construct is less a single proposal than a line of work. The reviewer's suggestions such as consistency, calibration, and distribution sensitivity are promising, and calibration/consistency are especially attractive because of the guarantees they might imply for users. Our main point is that future work should propose such constructs explicitly, identify candidate indicators, and then rigorously test whether they measure something stable, useful, and externally relevant.
>
>
> We appreciate this question and the feedback and agree that the current manuscript is not clear enough on why proposing constructs would exceed the scope of this work. We will make it more explicit in Section 6.
>
> **Q3-How to use existing benchmarks in the interim:** We recommend ceasing to interpret them as measures of human traits, before providing sufficient evidence for validity. We do not recommend discontinuing the use of human-inspired test items altogether. Instruments like IQ tests, personality inventories, bar exams, or medical licensing exams carry strong socio-cultural or institutional meaning and therefore create higher risk when aggregate scores are reported, even when used for engineering purposes only. Large-scale benchmarks inspired by human tests, such as MMLU, are different: they can still be useful for model ranking and comparison. Our main objection is to reporting aggregate scores from any of these instruments as measurements of human-like traits without the necessary validation of their measurement properties.
>
> [1] Burnell, Ryan, et al. "Revealing the structure of language model capabilities." arXiv preprint arXiv:2306.10062 (2023).

---

> > ### Author Rebuttal · Reviewer_YeVz · 2026-04-02
> >
> > I don't have other questions and will maintain my score.

---

### Official Review · Reviewer_bYyX · 2026-03-11

**Significance:** 4
**Argument Clarity:** 4
**Rating:** 6
**Confidence:** 4

**Questions:**

Minor suggestions for improvement (for additional questions see Weaknesses above)
* Figure 1: text in the subfigures is too small to be read in print
* L035-040l: sentence too long.
* L068l: potential typo "equivalently"?
* L069r: remove full stop after the quote for better typography
* Wrong quotes on L100r, L298-320l
* L229-230r, "English" should be capitalized
* L421-422l, author and citation should be in-text rather than repeated

**Alternative Views Section:**

Yes

**Compliance With Llm Reviewing Policy A Conservative:**

Affirmed.

**Discussion Potential:**

4

**Final Justification:**

The position is clearly stated, well-argued and is thought-provoking—and therefore very likely to inspire debate in the community.
I found the authors also meaningfully engaged with all the reviewers' comments through the rebuttal period, demonstrating a deep understanding of the literature behind the tools they discuss and this paper within the broader context of the field.

**I would like to also draw the meta-reviewer's attention** to the fact that the authors **have** meaningfully addressed Reviewer SCr2's critical comments and concerns, which Reviewer SCr2 subsequently chose to ignore in their final justification. I actually find the scope of the central claim to be very clear (i.e. that the field should not conflate the _numbers_ given by a benchmark with general claims about models' capabilities, like having specific psychological traits, "reasoning", "having PhD-level abilities", "consciousness", and other traits hyped up by the news cycles (!). Contrary to Reviewer SCr2, I would also say that a single in-depth case study with psychometric testing is _sufficient_ to prove that generalizing from benchmark performance to human traits is _not_ valid, while the onus should be on the _rest of the field_ to prove that such generalizations _are_ valid in other cases if at all (and that the field should not use misleading terminology in the meantime). In any case, this shows all the more that the paper has a lot of potential for debate, which is the whole goal of the paper as long as it is well-written and well-argued.

**Paper Summary:**

The authors propose the idea that many tests for evaluating human qualities (e.g. personality tests) were designed with specific (human) populations in mind, and therefore cannot meaningfully carry over to LLM evaluation, resulting in misleading claims and perceptions. In addition to not actually measuring the same underlying qualities, they argue that such findings are not falsifiable and may be confounded by alternative explanations. A call to action is made to develop new, AI-specific benchmarking standards, and to make it clear what exact, AI-specific constructs are to be measured without conflating them to human ones.

**Position:**

Yes

**Position In Title:**

Yes

**Related Work:**

4

**Strengths And Weaknesses:**

Strengths:
* Clear position statement, very clear, easy-to-read introduction
* Section 2 provides a comprehensive overview of the related events and perspective shifts that motivated the position, and how similar problems have been identified by other individual works. There is a clear reasoning and explanation for how the current perspective developed
* Section 3 provides a clear argument for why the authors believe the psychological tests were not designed to evaluate LLMs. The exact error that the authors disagree with and motivate the position by (e.g. orange highlights in Figure 1, Section 3.2) is identified and put in the broader context.
* A well-reasoned explanation is provided why previous research can be misleading (Section 4) and why the authors think the current research methods are incorrect, encouraging constructive debate and discussion in the community.
* The position has a level of nuance, e.g. discussed benefits _and_ drawbacks of their suggestion to use separate concepts for human vs LLM testing (e.g. overloading language)
* An argument is given for why debate and consideration is actually important by discussing the consequences and impacts (Section 5), but is also framed positively and accessibly as an opportunity (Section 6)
* Specific, concrete questions and calls to action are given
* Alternative philosophies (e.g. Popper vs Feyerabend) are considered, with concrete citations of relevant work

Weaknesses
* The alternative view discussion could be strengthened further. In particular, in L415l "these findings neither support broader claims about benchmarks..." could be seen to contradict the previous "anything goes"/"Against method" philosophy, where even an "invalid" benchmark should be considered seriously in order to allow for the progress of science as a whole. In other words, if methodological pluralism and anarchy is considered seriously, then alternative benchmarking strategies should be welcomed and encouraged, even if they seem "incorrect" according to the current standards. It is also unclear what do "original" vs "recreated" test sets have to do with the question at hand (i.e., applying the _same_ human test to LLMs) so authors could elaborate.
* That same section could also emphasize a bit better the distinction of what the authors mentioned earlier in the paper, the conflation of benchmark as a tool to measure "accuracy on a task" and benchmark as a "tool to measure a certain implicit quality", as it was not very easy to follow.

**Support:**

4

---

> ### Author Rebuttal · Authors · 2026-03-30
>
> We thank Reviewer bYyX for their positive and detailed review.
>
> **Alternative views and methodological pluralism:** We agree this is a helpful suggestion for strengthening the paper. Our view is only partially Feyerabendian: we welcome new methods, but within the iron rule of ML research, i.e. methods should earn their place by producing robust, generalizable improvements. In that sense, these works have clearly been welcomed by the field: they have appeared at top venues and been widely cited. The problem begins when the inference shifts from dataset performance to measurement of human-like traits. Under that stronger interpretation, aggregate accuracy scores can neither confirm nor falsify the trait claim, so the method no longer supports the kind of cumulative progress the iron rule relies on. Researchers may eventually notice that such claims fail to generalize, but in the meantime the unsupported human-trait inferences themselves can impose public harms and misallocated effort. We will make this distinction much clearer in the revision.
>
> **Original vs. recreated test sets:** We agree that this connection could be clearer. Our use of Recht et al. (2019) is not to argue directly about applying human tests to LLMs, but to make a narrower point about interpretation: benchmark *rankings* can be robust even when specific *accuracy scores* are not. This matters because many human-test comparisons with LLMs are communicated through exact score interpretations. We will revise this section to make that connection explicit.
>
> **Benchmarking vs. measurement distinction in Section 7:** We agree this could be stated more clearly. As discussed in our response to Reviewer SCr2, we will add a dedicated sentence to the introduction foregrounding the benchmarking/measurement distinction.
>
> **Minor corrections:** Thank you for the careful proofreading. We will fix all noted typos, formatting issues, and the figure readability in the revision.

---

> > ### Author Rebuttal · Reviewer_bYyX · 2026-04-02
> >
> > Thanks for your response.
> >
> > Regarding the Feyerabendian view, I understand you agree to it only partially, but the "Alternative View" itself could be to discuss Feyerabendianism as-is (rather than just the extension by Hardt and others), which would consider comparing raw accuracies (and even potentially drawing conclusions about human-like traits) acceptable. For example Feyerabend's work itself doesn't support the "provided it improves performance on the benchmark" clause you mention in L398. (I would also perhaps cite the original book from 1970s rather than the 2020 reprinting way after the author's death).
> > I am fine with the rest of the argument and appreciate the revision.
> >
> > I also realize my original questions might have not made much sense as Section 7 in particular was pretty confusing to me on that reading (even though on this reading I understand what was meant better). So I would indeed appreciate if somehow that part of the paper was simplified and the distinction between 1) ranking, 2) measuring individual accuracies, 3) drawing conclusions about implicit human traits, was emphasized throughout the section; maybe they could be somehow less entangled to make it easier to track those distinctions. For example, including a reminder in L417 that "measurement instruments" now relate to the conclusions about implicit traits rather than the measurement of individual accuracies would be helpful for readers not having psychology terms as built-in in their working memory, etc.
> >
> > Otherwise my comments have been fully resolved.

---

### Official Review · Reviewer_SCr2 · 2026-03-13

**Significance:** 2
**Argument Clarity:** 3
**Rating:** 2
**Confidence:** 5

**Questions:**

See Weakness above

**Alternative Views Section:**

Yes

**Compliance With Llm Reviewing Policy A Conservative:**

Affirmed.

**Discussion Potential:**

2

**Final Justification:**

Thank you for the clarification. After re-reading the paper and other reviewers' comments, I believe my original concern remains: the central claim is broader than the direct evidence provided. The argument is strongest when discussing trait-style human tests, but the paper repeatedly extends this critique to benchmark-based measurement claims more generally, while the empirical illustration remains narrow and based on one specific personality-test setup.

More importantly, there is a substantial body of prior work on using psychometric and human-testing frameworks for AI evaluation that is not meaningfully discussed or cited. Some of this work even takes a position directly opposed to the one advanced here. For a position paper, this omission is especially problematic, because it weakens the sense that the paper is engaging seriously and comprehensively with the existing debate.

[1] tinybenchmarks: evaluating LLMs with fewer examples. ICML, 2024.

[2] Position: AI Evaluation Should Learn from How We Test Humans. ICML, 2025

[3] metabench–a sparse benchmark to measure general ability in large language models. arXiv preprint arXiv:2407.12844, 2024.

[4] Building an evaluation scale using item response theory. In Proceedings of the Conference on Empirical Methods in Natural Language Processing. Conference on Empirical Methods in Natural Language Processing, volume 2016

[5] On evaluation of vision datasets and models using human competency frameworks. arXiv preprint arXiv:2409.04041, 2024.

I agree the paper raises an interesting and important issue, but in its current form the evidence and framing still do not fully support the breadth of the conclusion, so I am maintaining my original score.

**Paper Summary:**

The article argues that using intelligence, personality, or academic tests originally designed for humans to evaluate large language models is a category mistake. The authors stress that psychological and educational tests are measurement tools calibrated for specific human populations, and applying them directly to AI without validation leads to misleading interpretations because it lacks both theoretical and empirical justification. Current evaluation practices are widely undermined by limited validity, data contamination, cultural bias, and excessive sensitivity to prompt wording. In experiments, LLMs often also fail to meet the psychometric requirement of measurement invariance, meaning that the same test may not be measuring the same construct in humans and AI. This anthropomorphic approach to evaluation can mislead the public, foster unwarranted trust, and even allow model developers to evade legal responsibility by invoking ideas such as “model personality.” The paper therefore calls for an end to the use of human tests for AI evaluation. Instead, it advocates drawing on the rigor of psychometrics and econometrics, while leveraging AI’s unique capacity for direct causal investigation, to build dedicated evaluation frameworks grounded in AI-specific behavioral dimensions and supported by a strong theoretical foundation.

**Position:**

Yes

**Position In Title:**

Yes

**Related Work:**

2

**Strengths And Weaknesses:**

Strengths:
1. The paper moves beyond intuitive criticism by introducing established psychometric concepts such as Measurement Invariance and Construct Validity , providing a formal mathematical framework for machine learning evaluation through the lens of Structural Equation Models (SEMs).

2. The authors propose four specific open problems for the community to resolve, such as defining the proper statistical unit of measurement and identifying representative samples for the population of LLMs. These challenges provide a roadmap for developing a future science of evaluation for AI.

Weakness:

1. The core empirical evidence is based almost entirely on personality inventories, which are designed to measure highly subjective and socially-constructed latent traits. It is a significant leap in logic to generalize the failure of these specific instruments to all AI benchmarks, including those measuring objective academic knowledge or logic.

2. The paper dismisses the validity of performance-based scoring on functional tests. However, the ability to solve a specific task (e.g., mathematical reasoning or coding) is distinct from the possession of a latent psychological trait, and the paper fails to demonstrate why the former should be invalidated by the absence of the latter.

3. The critique fails to maintain a clear distinction between engineering utility (benchmarking) for model ranking and scientific measurement of latent traits. While the authors acknowledge that benchmarks like MMLU serve for healthy competition and model comparison , they simultaneously criticize them for failing to meet the strict standards of psychometric measurement models.

4. The experimental results are limited to a single model version (gpt-4-0613). Furthermore, the methodology of mapping "refusals" to a "Neutral" response category may artificially distort the statistical variance and factor loadings, potentially biasing the conclusions regarding the failure of measurement invariance.

5. The robot heart rate monitor analogy neglects the fundamental difference between functional outputs and biological traits. While a pulse is a physiological byproduct, standardized tests like the GRE measure information processing results that can be objectively compared across different systems, regardless of whether their internal mechanisms (carbon vs. silicon) are the same.

6. Measurement invariance is typically used to ensure that test items function similarly across different human subgroups, such as different cultures or languages. Demanding that completely different architectures (LLMs and humans) maintain identical factor loadings effectively mandates that an AI must mimic human cognitive structures to be considered evaluable, which may be an unnecessarily restrictive barrier for scientific cross-comparison.

**Support:**

1

---

> ### Author Rebuttal · Authors · 2026-03-30
>
> We thank Reviewer SCr2 for the detailed review. Several criticisms rest on positions our paper does not hold or on factual claims contradicted by the literature we cite. We clarify each point below and indicate where we will revise the manuscript.
>
> **W1)** We do not claim a failure of all AI benchmarks. Benchmarks work well for evaluation and driving innovation, as we explicitly state in Sections 4.2 L 257: “Note that this critique is only directed towards interpreting benchmarks as measurements. A leaderboard on a benchmark like MMLU still serves the purpose of model comparison and healthy competition over the best performance…” and reiterate in Sections 7 and 8.
>
> The reviewer argues that generalizing from personality to benchmarks measuring "objective academic knowledge or logic" is a significant leap. This misreads both our evidence base and our argument's direction. On evidence: we cite Jung et al. (2026) for morality and racism scales, McCoy et al. (2023) showing performance is driven by output probability rather than latent abilities, and Schröder et al. (2025) for instability in cognitive tests. On direction: personality is our working example precisely because its theoretical commitments are weaker. Constructs like intelligence or reasoning rest on far more elaborate latent-variable theories, each carrying stronger assumptions. The reviewer's contrast between "subjective" personality and "objective" academic knowledge is contested, since both reflect theoretical choices made by and validated on human populations.
>
> **W2)** The reviewer states the paper "fails to demonstrate why [task performance] should be invalidated by the absence of [latent psychological traits]." We do not invalidate task-level accuracy scores (Section 4.2, 7, 8). Our critique targets only the interpretive leap from task score to latent trait claim. We note in Sections 2 and 5 that purely functional use of *human* tests invite this leap.
>
> **W3)** The distinction between benchmarking and measurement is the paper's central organizing argument. It is drawn in Section 2 (documenting the historical shift in language from "dataset" to "measures intelligence"), stated explicitly in Section 4.2 and developed at length in Section 7, which is entirely dedicated to affirming the value of benchmarking as a driver of progress and restated in Section 8. Since this was not perceived as clear we will add a dedicated sentence to the introduction for clarity.
>
> **W4)** Appendix A, while an illustrative example, is not our core evidence. Beyond our theoretical arguments (Meredith, 1993, Cronbach & Meehl 1955), Sühr et al. (2023) tested multiple prompt settings and models, none showing measurement invariance. Additional empirical evidence comes from Schröder et al. (2025), Jung et al. (2026), and McCoy et al. (2023).
>
> **W5)** The reviewer argues GRE scores "can be compared across different systems regardless of internal mechanisms." This is contradicted by ETS's own practices: ETS routinely conducts differential item functioning and measurement invariance studies before authorizing cross-group comparisons, even within the human population [1,2]. These tests are agnostic about how individuals arrive at answers, but they are normed on a distribution of inter-individual differences to support calibrated predictions within that population (people with score x have probability of academic success y). GRE item difficulty is a population-relative parameter, not an intrinsic property. LLMs may find human-hard items trivial while failing human-easy ones (how many r's in strawberry), showing that inter-individual comparisons do not generalize across systems. The heart rate analogy is apt: the problem is not obtaining a number, but assuming it carries the inter-individual interpretation it was calibrated to carry.
>
> **W6)** Strong measurement invariance is not a bar we impose on AI evaluation; it is a mathematical requirement (Meredith, 1993) for score comparisons across populations. We are not demanding that LLMs mimic human cognition, we point out that if they do not, then scores cannot be compared as measurements of the same latent trait. This is a logical consequence, not a normative restriction. The reviewer's observation that LLMs and humans may have fundamentally different architectures is therefore not a counterargument, it is a reason to expect invariance to fail, which supports our position. Weaker forms of invariance permit different and more limited conclusions, which could inform the discussion within the ICML community. We will add an explicit discussion of them in the revision.
>
> [1]Rock, D. A., Bennett, R. E., & Jirele, T. (1986). The internal construct validity of the GRE general test across handicapped and nonhandicapped populations. ETS Research Report Series, 1986.
>
> [2]Broer, M. (2005). Ensuring the fairness of GRE writing prompts: Assessing differential difficulty. ETS Research Report Series, 2005.

---

> > ### Author Rebuttal · Reviewer_SCr2 · 2026-04-03
> >
> > Thank you for the clarification.  After re-reading the paper and other reviewers' comments, I believe my original concern remains: the central claim is broader than the direct evidence provided. The argument is strongest when discussing trait-style human tests, but the paper repeatedly extends this critique to benchmark-based measurement claims more generally, while the empirical illustration remains narrow and based on one specific personality-test setup.
> >
> > More importantly, there is a substantial body of prior work on using psychometric and human-testing frameworks for AI evaluation that is not meaningfully discussed or cited. **Some of this work even takes a position directly opposed to the one advanced here**. For a position paper, this omission is especially problematic, because it weakens the sense that the paper is engaging seriously and comprehensively with the existing debate.
> >
> > [1] tinybenchmarks: evaluating LLMs with fewer examples. ICML, 2024.
> >
> > [2] Position: AI Evaluation Should Learn from How We Test Humans. ICML, 2025
> >
> > [3] metabench–a sparse benchmark to measure general ability in large language models. arXiv preprint arXiv:2407.12844, 2024.
> >
> > [4] Building an evaluation scale using item response theory. In Proceedings of the Conference on Empirical Methods in Natural Language Processing. Conference on Empirical Methods in Natural Language Processing, volume 2016
> >
> > [5] On evaluation of vision datasets and models using human competency frameworks. arXiv preprint arXiv:2409.04041, 2024.
> >
> > I agree the paper raises an interesting and important issue, but in its current form the evidence and framing still do not fully support the breadth of the conclusion, so I am maintaining my original score.

---

### Official Review · Reviewer_hVwE · 2026-03-16

**Significance:** 4
**Argument Clarity:** 3
**Rating:** 5
**Confidence:** 3

**Questions:**

- Section 4.2: Is it fundamentally possible to establish measurement invariance between LLMs and humans? What are examples of latent capabilities that *could* be shared between LLMs and humans?
- How does the sampling variability of LLMs (as opposed to humans) affect the feasibility of designing measurement instruments for latent LLM characteristics?
- Section 4.4: How might standards for validating LLM benchmarks be governed?

**Alternative Views Section:**

Yes

**Compliance With Llm Reviewing Policy A Conservative:**

Affirmed.

**Discussion Potential:**

4

**Final Justification:**

The authors have addressed my weaknesses and questions in detail with their promised revisions. I would like to maintain my score as is.

**Paper Summary:**

- The paper argues that LLMs should not be assessed using psychological and educational tests that are designed for humans without more rigorous validation, as doing so can produce miscalibrated results (population mismatch) or measure meaningless constructs (construct mismatch).
- The paper makes this argument by discussing: (1) how to rigorously construct and test measurement instruments, (2) central properties of measurements, (3) the dangers of conveying invalid measurements to the public that suggest that LLMs have human-like traits, and (4) concrete research directions towards building valid measurement instruments for LLMs.

**Position:**

Yes

**Position In Title:**

Yes

**Related Work:**

3

**Strengths And Weaknesses:**

*Strengths:*
- The paper will inspire important discussions about: (1) how using tests designed for humans to evaluate LLMs can lead to invalidly attributing human-like traits to LLMs, and (2) research directions to establish a science of measurement for LLMs. The paper provides numerous thoughtful research directions (Section 6), e.g., at what level should LLMs be evaluated, and how can we build principled measurement tools that do not depend on a specific population of LLMs?
- The paper provides detailed background information about how claims of what benchmarks assess have changed over time, how human psychological and educational tests have been applied to LLMs, and measurement invariance.
- The paper clearly deconstructs how human educational and psychological measurement instruments are developed and why applying them to LLMs produces invalid measurements. Figure 1 is a clear and helpful diagram of the process we would need to follow to develop valid measurement tools for LLMs.
- The paper clearly argues why using human measurement instruments to test if LLMs are human-like is flawed through the lenses of experimental risk and measurement invariance. The commentary (Section 4.2) on the ambiguous nature of measurement invariance between models that are finetuned on more vs. less relevant data for a benchmark is interesting.

*Weaknesses:*
- In Section 2, the paper should elaborate on how their arguments differ from prior work that critically analyzes the measurement validity of benchmarks.
- The paper could more explicitly connect its arguments to prior work on measurement validity by, e.g., discussing convergent and hypothesis validity in Section 4.3 [1].
- Regarding alternative views, the paper should address criticism of how psychological tests often do not have measurement invariance for humans (e.g., across cultural backgrounds).

[1] Jacobs, A. Z., & Wallach, H. (2021, March). Measurement and fairness. In Proceedings of the 2021 ACM conference on fairness, accountability, and transparency (pp. 375-385).

**Support:**

4

---

> ### Author Rebuttal · Authors · 2026-03-30
>
> We thank Reviewer hVwE for the thoughtful review and helpful suggestion. We agree that the paper should better clarify its contribution relative to prior critical work. To address this, we will add the following paragraph at the end of Section 2:
>
> **Differentiating from prior work (Section 2):** Thank you for pointing out that we can better articulate our distinct contribution relative to prior critical work. In the end of Section 2, we will add the following paragraph:
>
> *While these works each identify specific failures empirically, this paper provides a unified theoretical account of what would constitute sufficient evidence to support a measurement claim, grounding the criteria for validity in measurement invariance, structural equation models, and nomological networks, and offers an end-to-end framework for constructing and validating measurement instruments that meet these criteria.*
>
> *Distinctively, we argue, following Popper (1989) and Meehl (1992), that current evaluation designs are structurally too weak to support measurement claims, as they rarely place the underlying measurement hypothesis at genuine risk of refutation.*
>
> We thank the reviewer for highlighting this point; this revision will make our distinct contribution clearer.
>
> **Connecting to convergent and hypothesis validity:** Thank you for this helpful suggestion. Jacobs & Wallach (2021) provide a valuable framework that complements our argument, particularly with respect to construct validity in fairness-related measurement. We will discuss this work in more detail in the related work section and also connect it to our discussion of nomological networks in Section 4.3. To clarify this connection, we will add the following paragraph:
>
> *This issue directly connects to the concepts of convergent and discriminant validity (Jacobs & Wallach 2021): a benchmark claiming to measure 'general intelligence' should not only correlate with other benchmarks making similar claims (convergent validity), but should also fail to correlate with benchmarks measuring demonstrably distinct constructs (discriminant validity). Without both, correlation within a nomological network provides weak evidential support.*
>
> **Measurement invariance failures across human cultures:** This is a great suggestion. We will add a discussion to the alternative views section explaining how psychologists fall back on weaker forms of invariance (e.g., configural invariance) where parameter values can differ to some degree while structural equality is required. Strong measurement invariance is the requirement for score comparisons between populations, but weaker forms of invariance allow for other comparisons, such as similarities of the latent variable model (e.g. in the case of configural invariance). However, neither the works we cite nor our empirical analysis suggests even the weakest forms of invariance between humans and LLMs. We will also add an overview of weaker forms of invariance and the level of interpretation they allow for.
>
> **Questions:**
> - *Is measurement invariance between LLMs and humans fundamentally possible?* Yes, models can be trained to satisfy measurement invariance (adopting the perspective of models as populations from Section 6). However, establishing *validity* requires the process we outline in Figure 1 and other qualitative discussions as outlined in the Standards of Educational and Psychological Testing.
> - *Shared latent capabilities:* Identifying shared constructs is an important open problem in its own right (Section 6). Proposing plausible candidates is a natural starting point, but demonstrating that a construct is truly shared requires the full validation process we outline, which is a substantial undertaking. We therefore frame this as an exciting research direction rather than a solved question.
> - *Sampling variability of LLMs:* LLMs offer unique advantages, in human psychology, multi-trait-multi-method approaches try to disentangle trait variance from method variance. Test-retest variance in humans is extremely hard to disentangle since we do not know what the source of the variation is. In LLMs, for open-weight models, we can quantify the variance for each item precisely. We can easily distinguish between trait and method variance and there should not be any additional test-retest variance beyond that.
> - *Governance of standards:* We envision a community-driven process, analogous to the Standards for Educational and Psychological Testing. In psychology, organizations like the APA coordinate dedicated working groups and task forces for specific constructs (e.g., intelligence, personality), which iteratively develop and revise testing standards. A similar model could be adopted in ML, coordinated through venues like ICML, NeurIPS, or dedicated workshops, with working groups for specific evaluation domains. We discuss this briefly in Section 4.4 but are happy to add more suggestions in the camera ready version.

---

> > ### Author Rebuttal · Reviewer_hVwE · 2026-04-03
> >
> > Thanks for your detailed and thoughtful response! I appreciate the inclusion of these revisions in the next draft and the clarifications and responses to my questions. I will maintain my score as is.

---

### Decision · Program_Chairs · 2026-04-30

**Decision:**

Accept (regular)

**Comment:**

I recommend acceptance. The paper raises a timely and important methodological issue, is clearly argued, and appears to have generated exactly the kind of substantive debate the position-paper track is meant to encourage; moreover, the review set is overall positive after rebuttal, with three supportive reviewers maintaining accept-level scores and indicating that their concerns were fully resolved, while the remaining negative review centers largely on scope and related-work framing rather than on a failure to articulate a legitimate position.